# Bitrate-Constrained DRO: Beyond Worst Case Robustness To Unknown Group Shifts

**Amrith Setlur**[1]    **Don Dennis**[1]    **Benjamin Eysenbach**[1]
**Aditi Raghunathan**[1]    **Chelsea Finn**[2]    **Virginia Smith**[1]    **Sergey Levine**[3]

[1] Carnegie Mellon University    [2] Stanford University    [3] UC Berkeley

## Abstract

Training machine learning models robust to distribution shifts is critical for real-world applications. Some robust training algorithms (*e.g.,* Group DRO) specialize to group shifts and require group information on all training points. Other methods (*e.g.,* CVaR DRO) that do not need group annotations can be overly conservative, since they naively upweight high loss points which may form a contrived set that does not correspond to any meaningful group in the real world (*e.g.,* when the high loss points are randomly mislabeled training points). In this work, we address limitations in prior approaches by assuming a more nuanced form of group shift: conditioned on the label, we assume that the true group function (indicator over group) is simple. For example, we may expect that group shifts occur along low bitrate features (*e.g.,* image background, lighting). Thus, we aim to learn a model that maintains high accuracy on simple group functions realized by these low bitrate features, that need not spend valuable model capacity achieving high accuracy on contrived groups of examples. Based on this, we consider the two-player game formulation of DRO where the adversary's capacity is bitrate-constrained. Our resulting practical algorithm, Bitrate-Constrained DRO (`BR-DRO`), does not require group information on training samples yet matches the performance of Group DRO on datasets that have training group annotations and that of CVaR DRO on long-tailed distributions. Our theoretical analysis reveals that in some settings `BR-DRO` objective can provably yield statistically efficient and less conservative solutions than unconstrained CVaR DRO.

## 1 Introduction

Machine learning models may perform poorly when tested on distributions that differ from the training distribution. A common form of distribution shift is *group* shift, where the source and target differ only in the marginal distribution over finite groups or sub-populations, with no change in group conditionals (Oren et al., 2019; Duchi et al., 2019) (*e.g.,* when the groups are defined by spurious correlations and the target distribution upsamples the group where the correlation is absent Sagawa et al. (2019)).

Prior works consider various approaches to address group shift. One solution is to ensure robustness to worst case shifts using distributionally robust optimization (DRO) (Bagnell, 2005; Ben-Tal et al., 2013; Duchi et al., 2016), which considers a two-player game where a learner minimizes risk on distributions chosen by an adversary from a predefined uncertainty set. As the adversary is only constrained to propose distributions that lie within an f-divergence based uncertainty set, DRO often yields overly conservative (pessimistic) solutions (Hu et al., 2018) and can suffer from statistical challenges (Duchi et al., 2019). This is mainly because DRO upweights high loss points that may not form a meaningful group in the real world, and may even be *contrived* if the high loss points simply correspond to randomly mislabeled examples in the training set. Methods like Group DRO (Sagawa et al., 2019) avoid overly pessimistic solutions by assuming knowledge of group membership for each training example. However, these group-based methods provide no guarantees on shifts that deviate from the predefined groups (*e.g.,* when there is a new group), and are not applicable to problems that lack group knowledge. In this work, we therefore ask: *Can we train non-pessimistic robust models without access to group information on training samples?*

We address this question by considering a more nuanced assumption on the structure of the underlying groups. We assume that, conditioned on the label, group boundaries are realized by high-level features that depend on a small set of underlying factors (*e.g.,* background color, brightness). This leads to simpler group

---

*Correspondence can be sent to asetlur@cs.cmu.edu.

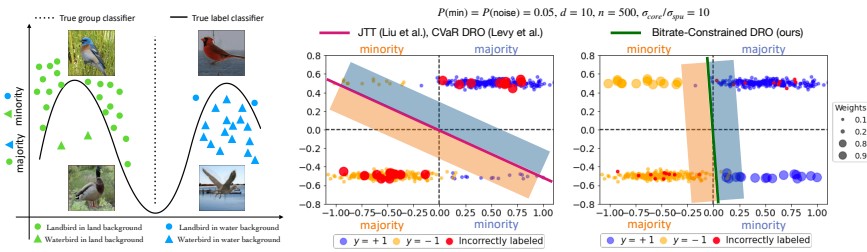

Figure 1: **Bitrate-Constrained DRO**: A method that assumes group shifts along low-bitrate features, and restricts the adversary appropriately so that the solution found is less pessimistic and more robust to unknown group shifts. Our method is also robust to training noise. *(Left)* In Waterbirds (Wah et al., 2011), the spurious feature background is a large margin simple feature that separates the *majority* and *minority* points in each class. *(Right)* Prior works (Levy et al., 2020; Liu et al., 2021) that upweight arbitrary points with high losses force the model to memorize noisy mislabeled points while our method is robust to noise and only upweights the true minority group without any knowledge of its identity (see Section 6.2).

functions with large margin and simple decision boundaries between groups (Figure 1 *(left)*). Invoking the principle of minimum description length (Grünwald, 2007), restricting our adversary to functions that satisfy this assumption corresponds to a bitrate constraint. In DRO, the adversary upweights points with higher losses under the current learner, which in practice often correspond to examples that belong to a rare group, contain complex patterns, or are mislabeled (Carlini et al., 2019; Toneva et al., 2018). Restricting the adversary's capacity prevents it from upweighting individual hard or mislabeled examples (as they cannot be identified with simple features), and biases it towards identifying erroneous data points misclassified by simple features. This also complements the failure mode of neural networks trained with stochastic gradient descent (SGD) that rely on simple spurious features which correctly classify points in the *majority* group but may fail on *minority* groups (Blodgett et al., 2016).

The main contribution of this paper is Bitrate-Constrained DRO (`BR-DRO`), a supervised learning procedure that provides robustness to distribution shifts along groups realized by simple functions. Despite not using group information on training examples, we demonstrate that `BR-DRO` can match the performance of methods requiring them. We also find that `BR-DRO` is more successful in identifying true minority training points, compared to unconstrained DRO. This indicates that not optimizing for performance on contrived worst-case shifts can reduce the pessimism inherent in DRO. It further validates: (i) our assumption on the simple nature of group shift; and (ii) that our bitrate constraint meaningfully structures the uncertainty set to be robust to such shifts. As a consequence of the constraint, we also find that `BR-DRO` is robust to random noise in the training data (Song et al., 2022), since it cannot form "groups" entirely based on randomly mislabeled points with low bitrate features. This is in contrast with existing methods that use the learner's training error to up-weight arbitrary sets of difficult training points (*e.g.,* Liu et al., 2021; Levy et al., 2020), which we show are highly susceptible to label noise (see Figure 1 *(right)*). Finally, we theoretically analyze our approach—characterizing how the degree of constraint on the adversary can effect worst risk estimation and excess risk (pessimism) bounds, as well as convergence rates for specific online solvers.

## 2 RELATED WORK

Prior works in robust ML (e.g., Li et al., 2018; Lipton et al., 2018; Goodfellow et al., 2014) address various forms of adversarial or structured shifts. We specifically review prior work on robustness to group shifts. While those based on DRO optimize for worst-case shifts in an explicit uncertainty set, the robust set is implicit for some others, with most using some form of importance weighting.

**Distributionally robust optimization (DRO).** DRO methods generally optimize for worst-case performance on joint $(\mathbf{x},\mathbf{y})$ distributions that lie in an $f$-divergence ball (uncertainty set) around the training distribution (Ben-Tal et al., 2013; Rahimian & Mehrotra, 2019; Bertsimas et al., 2018; Blanchet & Murthy, 2019; Miyato et al., 2018; Duchi et al., 2016; Duchi & Namkoong, 2021). Hu et al. (2018) highlights that the conservative nature of DRO may lead to degenerate solutions when the unrestricted adversary uniformly upweights all misclassified points. Sagawa et al. (2019) proposes to address this by limiting the adversary to shifts that only differ in marginals over predefined groups. However, in addition to it being difficult to obtain this information, Kearns et al. (2018) raise "gerrymandering" concerns with notions of robustness that fix a small number of groups apriori. While they propose a solution that looks at exponentially many subgroups defined over protected attributes, our method does not assume access to such attributes and

aims to be fair on them as long as they are realized by simple functions. Finally, Zhai et al. (2021) avoid conservative solutions by solving the DRO objective over randomized predictors learned through boosting. We consider deterministic and over-parameterized learners and instead constrain the adversary's class.

**Constraining the DRO uncertainty set.** In the marginal DRO setting, Duchi et al. (2019) limit the adversary via easier-to-control reproducing kernel hilbert spaces (RKHS) or bounded Hölder continuous functions (Liu & Ziebart, 2014; Wen et al., 2014). While this reduces the statistical error in worst risk estimation, the size of the uncertainty set (scales with the data) remains too large to avoid cases where an adversary can re-weight mislabeled and hard examples from the majority set (Carlini et al., 2019). In contrast, we restrict the adversary even for large datasets where the estimation error would be low, as this would reduce excess risk when we only care about robustness to rare sub-populations defined by simple functions. Additionally, while their analysis and method prefers the adversary's objective to have a strong dual, we show empirical results on real-world datasets and generalization bounds where the adversary's objective is not necessarily convex.

**Robustness to group shifts without demographics.** Recent works (Sohoni et al., 2020; Creager et al., 2021; Bao & Barzilay, 2022) that aim to achieve group robustness without access to group labels employ various heuristics where the robust set is implicit while others require data from multiple domains (Arjovsky et al., 2019; Yao et al., 2022) or ability to query test samples (Lee et al., 2022). Liu et al. (2021) use training losses for a heavily regularized model trained with empirical risk minimization (ERM) to directly identify minority data points with higher losses and re-train on the dataset that up-weights the identified set. Nam et al. (2020) take a similar approach. Other methods (Idrissi et al., 2022) propose simple baselines that subsample the majority class in the absence of group demographics and the majority group in its presence. Hashimoto et al. (2018) find DRO over a $\chi^2$-divergence ball can reduce the otherwise increasing disparity of per-group risks in a dynamical system. Since it does not use features to upweight points (like `BR-DRO`) it is vulnerable to label noise. Same can be said about some other works (*e.g.,* Liu et al. (2021); Nam et al. (2020)).

**Importance weighting in deep learning.** Finally, numerous works (Duchi et al., 2016; Levy et al., 2020; Lipton et al., 2018; Oren et al., 2019) enforce robustness by re-weighting losses on individual data points. Recent investigations (Soudry et al., 2018; Byrd & Lipton, 2019; Lu et al., 2022) reveal that such objectives have little impact on the learned solution in interpolation regimes. One way to avoid this pitfall is to train with heavily regularized models (Sagawa et al., 2019; 2020) and employ early stopping. Another way is to subsample certain points, as opposed to up-weighting (Idrissi et al., 2022). In this work, we use both techniques while training our objective and the baselines, ensuring that the regularized class is robust to shifts under misspecification (Wen et al., 2014).

## 3 PRELIMINARIES

We introduce the notation we use in the rest of the paper and describe the DRO problem. In the following section, we will formalize our assumptions on the nature of the shift before introducing our optimization objective and algorithm.

**Notation.** With covariates $\mathcal{X} \subset \mathbb{R}^d$ and labels $\mathcal{Y}$, the given source $P$ and unknown true target $Q_0$ are measures over the measurable space $(\mathcal{X} \times \mathcal{Y}, \Sigma)$ and have densities $p$ and $q_0$ respectively (w.r.t. base measure $\mu$). The learner's choice is a hypothesis $h : \mathcal{X} \mapsto \mathcal{Y}$ in class $\mathcal{H} \subset L^2(P)$, and the adversary's action in standard DRO is a target distribution $Q$ in set $\mathcal{Q}_{P,\kappa} \coloneqq \{Q : Q \ll P, D_f(Q \| P) \leq \kappa\}$. Here, $D_f$ is the $f$-divergence between $Q$ and $P$ for a convex function $f$[1] with $f(1) = 0$. An equivalent action space for the adversary is the set of re-weighting functions:

$$\mathcal{W}_{P,\kappa} = \{w : \mathcal{X} \times \mathcal{Y} \mapsto \mathbb{R} : w \text{ is measurable under } P, \mathbb{E}_P[w] = 1, \mathbb{E}_P f(w) \leq \kappa\} \quad (1)$$

For a convex loss function $l : \mathcal{Y} \times \mathcal{Y} \mapsto \mathbb{R}_+$, we denote $l(h)$ as the function over $(\mathbf{x}, \mathbf{y})$ that evaluates $l(h(\mathbf{x}), \mathbf{y})$, and use $l_{0-1}$ to denote the loss function $\mathbb{1}(h(\mathbf{x}) \neq \mathbf{y})$. Given either distribution $Q \in \mathcal{Q}_{P,\kappa}$, or a re-weighting function $w \in \mathcal{W}_{P,\kappa}$, the risk of a learner $h$ is:

$$R(h, Q) = \mathbb{E}_Q[l(h)] \quad R(h, w) = \mathbb{E}_{(\mathbf{x}, \mathbf{y}) \sim P}[l(h(\mathbf{x}), \mathbf{y}) \cdot w(\mathbf{x}, \mathbf{y})] = \langle l(h), w \rangle_P \quad (2)$$

Note the overload of notation for $R(h, \cdot)$. If the adversary is stochastic it picks a mixed action $\delta \in \Delta(\mathcal{W}_{P,\kappa})$, which is the set of all distributions over $\mathcal{W}_{P,\kappa}$. Whenever it is clear, we drop $P, \kappa$.

**Unconstrained DRO (Ben-Tal et al., 2013).** This is a min-max optimization problem understood as a two-player game, where the learner chooses a hypothesis, to minimize risk on the worst distribution that

---

[1]For *e.g.,* $\text{KL}(Q \| P)$ can be derived with $f(x) = x \log x$ and for Total Variation $f(x) = |x - 1|/2$.

the adversary can choose from its set. Formally, this is given by Equation 3. The first equivalence is clear from the definitions and for the second since $R(h,Q)$ is linear in $Q$, the supremum over $\Delta(\mathcal{W}_{P,\kappa})$ is a Dirac delta over the best weighting in $\mathcal{W}_{P,\kappa}$. In the next section, we will see how a bitrate-constrained adversary can only pick certain actions from $\Delta(\mathcal{W}_{P,\kappa})$.

$$\inf_{h\in\mathcal{H}}\sup_{Q\in\mathcal{Q}_{P,\kappa}} R(h,Q) \equiv \inf_{h\in\mathcal{H}}\sup_{w\in\mathcal{W}_{P,\kappa}} R(h,w) \equiv \inf_{h\in\mathcal{H}}\sup_{\delta\in\Delta(\mathcal{W}_{P,\kappa})} \mathbb{E}_{w\sim\delta}[R(h,w)] \qquad (3)$$

**Group Shift.** While the DRO framework in Section 3 is broad and addresses any unstructured shift, we focus on the specific case of group shift. First, for a given pair of measures $P,Q$ we define what we mean by the group structure $\mathcal{G}_{P,Q}$ (Definition 3.1). Intuitively, it is a set of sub-populations along which the distribution shifts, defined in a way that makes them uniquely identifiable. For *e.g.,* in the Waterbirds dataset (Figure 1), there are four groups given by combinations of (label, background). Corollary 3.2 follows immediately from the definition of $\mathcal{G}_{P,Q}$. Using this definition, the standard group shift assumption (Sagawa et al., 2019) can be formally re-stated as Assumption 3.3.

**Definition 3.1** (group structure $\mathcal{G}_{P,Q}$). *For $Q \ll P$ the group structure $\mathcal{G}_{P,Q}=\{G_k\}_{k=1}^{K}$ is the smallest finite set of disjoint groups $\{G_k\}_{k=1}^{K}$ s.t. $Q(\cup_{k=1}^{K}G_k)=1$ and $\forall k$ (i) $G_k \in \Sigma$, $Q(G_k) > 0$ and (ii) $p(\mathbf{x},\mathbf{y}\,|\,G_k)=q(\mathbf{x},\mathbf{y}\,|\,G_k)>0$ a.e. in $\mu$. If such a structure exists then $\mathcal{G}_{P,Q}$ is well defined.*

**Corollary 3.2** (uniqueness of $\mathcal{G}_{P,Q}$). *$\forall P,Q$, the structure $\mathcal{G}(P,Q)$ is unique if it is well defined.*

**Assumption 3.3** (standard group shift). *There exists a well-defined group structure $\mathcal{G}_{P,Q_0}$ s.t. target $Q_0$ differs from $P$ only in terms of marginal probabilities over all $G\in\mathcal{G}_{P,Q_0}$.*

# 4 BITRATE-CONSTRAINED DRO

We begin with a note on the expressivity of the adversary in Unconstrained DRO and formally introduce the assumption we make on the nature of shift. Then, we build intuition for why unconstrained adversaries fail but restricted ones do better under our assumption. Finally, we state our main objective and discuss a specific instance of it.

**How expressive is unconstrained adversary?** Note that the set $\mathcal{W}_{P,\kappa}$ includes all measurable functions (under $P$) such that the re-weighted distribution is bounded in $f$-divergence (by $\kappa$). While prior works (Shafieezadeh Abadeh et al., 2015; Duchi et al., 2016) shrink $\kappa$ to construct confidence intervals, this *only controls* the total mass that can be moved between measurable sets $G_1,G_2 \in \Sigma$, but *does not restrict* the choice of $G_1$ and $G_2$ itself. As noted by Hu et al. (2018), such an adversary is highly expressive, and optimizing for the worst case only leads to the solution of empirical risk minimization (ERM) under $l_{0-1}$ loss. Thus, we can conclude that DRO recovers degenerate solutions because the worst target in $\mathcal{W}_{P,\kappa}$ lies far from the subspace of naturally occurring targets. Since it is hard to precisely characterize natural targets we make a nuanced assumption: the target $Q_0$ only upsamples those rare subpopulations that are misclassified by simple features. We state this formally in Assumption 4.2 after we define the bitrate-constrained function class $\mathcal{W}(\gamma)$ in Definition 4.1.

**Definition 4.1.** *A function class $\mathcal{W}(\gamma)$ is bitrate-constrained if there exists a data independent prior $\pi$, s.t. $\mathcal{W}(\gamma)=\{\mathbb{E}[\delta]:\delta\in\Delta(\mathcal{W}),KL(\delta\,\|\,\pi)\leq\gamma\}$.*

**Assumption 4.2** (simple group shift). *Target $Q_0$ satisfies Assumption 3.3 (group shift) w.r.t. source $P$. Additionally, For some prior $\pi$ and a small $\gamma^*$, the re-weighting function $q_0/p$ lies in a bitrate-constrained class $\mathcal{W}(\gamma^*)$. In other words, for every group $G\in\mathcal{G}(P,Q_0)$, $\exists w_G\in\mathcal{W}(\gamma^*)$ s.t. $\mathbb{1}((\mathbf{x},\mathbf{y})\in G)=w_G$ a.e.. We refer to such a $G$ as a **simple group** that is realized in $\mathcal{W}(\gamma^*)$.*

Under the principle of minimum description length (Grünwald, 2007) any deviation from the prior (*i.e.,* $KL(\delta\,\|\,\pi)$) increases the *description length* of the encoding $\delta\in\Delta(\mathcal{W})$, thus we refer to $\mathcal{W}(\gamma)$ as being *bitrate-constrained* in the sense that it contains functions (means of distributions) that can be described with a limited number of bits given the prior $\pi$. See Appendix A.3 for an example of a bitrate-constrained class of functions. Next we present arguments for why identifiability of simple (satisfy Assumption 4.2) minority groups can be critical for robustness.

**Neural networks can perform poorly on simple minorities.** For a fixed target $Q_0$, let's say there exists two groups: $G_{\min}$ and $G_{\maj}\in\mathcal{G}(P,Q_0)$ such that $P(G_{\min})\ll P(G_{\maj})$. By Assumption 4.2, both $G_{\min}$ and $G_{\maj}$ are simple (realized in $\mathcal{W}(\gamma^*)$), and are thus separated by some simple feature. The learner's class $\mathcal{H}$ is usually a class of overparameterized neural networks. When trained with stochastic gradient descent (SGD), these are biased towards learning simple features that classify a majority of the

data (Shah et al., 2020; Soudry et al., 2018). Thus, if the simple feature separating $G_{\text{min}}$ and $G_{\text{maj}}$ itself correlates with the label $y$ on $G_{\text{maj}}$, then neural networks would fit on this feature. This is precisely the case in the Waterbirds example, where the groups are defined by whether the simple feature background correlates with the label (Figure 1). Thus our assumption on the nature of shift complements the nature of neural networks perform poorly on simple minorities.

**The bitrate constraint helps identify simple unfair minorities in** $\mathcal{G}(P,Q_0)$**.** Any method that aims to be robust on $Q_0$ must up-weight data points from $G_{\text{min}}$ but without knowing its identity. Since the unconstrained adversary upsamples any group of data points with high loss and low probability, it cannot distinguish between a rare group that is realized by simple functions in $\mathcal{W}(\gamma^*)$ and a rare group of examples that share no feature in common or may even be mislabeled. On the other hand, the group of mislabeled examples cannot be separated from the rest by functions in $\mathcal{W}(\gamma^*)$. Thus, a bitrate constraint adversary can only identify simple groups and upsamples those that incur high losses – possibly due to the simplicity bias of neural networks.

**BR-DRO objective.** According to Assumption 4.2, there cannot exist a target $Q_0$ such that minority $G_{\text{min}} \in \mathcal{G}(P,Q_0)$ is not realized in bitrate constrained class $\mathcal{W}(\gamma^*)$. Thus, by constraining our adversary to a class $\mathcal{W}(\gamma)$ (for some $\gamma$ that is user defined), we can possibly evade issues emerging from optimizing for performance on mislabeled or hard examples, even if they were rare. This gives us the objective in Equation 4 where the equalities hold from the linearity of $\langle \cdot, \cdot \rangle$ and Definition 4.1.

$$\inf_{h \in \mathcal{H}} \sup_{\substack{\delta \in \Delta(\mathcal{W}) \\ \text{KL}(\delta \| \pi) \leq \gamma}} \mathbb{E}_{w \sim \delta} R(h,w) = \inf_{h \in \mathcal{H}} \sup_{\substack{\delta \in \Delta(\mathcal{W}) \\ \text{KL}(\delta \| \pi) \leq \gamma}} \langle l(h), \mathbb{E}_{\delta}[w] \rangle_P = \inf_{h \in \mathcal{H}} \sup_{w \in \mathcal{W}(\gamma)} R(h,w) \qquad (4)$$

**BR-DRO in practice.** We parameterize the learner $\boldsymbol{\theta}_h \in \Theta_h$ and adversary $\boldsymbol{\theta}_w \in \Theta_w$ as neural networks[2]. In practice, we implement the adversary either as a one hidden layer variational information bottleneck (VIB) (Alemi et al., 2016), where the Kullback-Leibler (KL) constraint on the latent variable $\mathbf{z}$ (output of VIB's hidden layer) directly constrains the bitrate; or as an $l_2$ norm constrained linear layer. The objective for the VIB ($l_2$) version is obtained by setting $\beta_{\text{vib}} \neq 0$ ($\beta_{l_2} \neq 0$) in Equation 5 below. See Appendix A.2 for details. Note that the objective in Equation 5 is no longer convex-concave and can have multiple local equilibria or stationary points (Mangoubi & Vishnoi, 2021). The adversary's objective also does not have a strong dual that can be solved through conic programs—a standard practice in DRO literature (Namkoong & Duchi, 2016). Thus, we provide an algorithm where both learner and adversary optimize BR-DRO iteratively through stochastic gradient ascent/descent (Algorithm 1 in Appendix A.1).

$$\min_{\boldsymbol{\theta}_h \in \Theta_h} \langle l(\boldsymbol{\theta}_h), \boldsymbol{\theta}_w^* \rangle_P \text{ s.t. } \boldsymbol{\theta}_w^* = \operatorname*{argmax}_{\boldsymbol{\theta}_w \in \Theta_w} L_{\text{adv}}(\boldsymbol{\theta}_w; \boldsymbol{\theta}_h, \beta_{\text{vib}}, \beta_{l_2}, \eta) \qquad (5)$$

$$L_{\text{adv}}(\boldsymbol{\theta}_w; \boldsymbol{\theta}_h, \beta_{\text{vib}}, \beta_{l_2}, \eta) = \langle l(\boldsymbol{\theta}_h) - \eta, \boldsymbol{\theta}_w \rangle_P - \beta_{\text{vib}} \mathbb{E}_P \text{KL}(p(\mathbf{z}|\mathbf{x}; \boldsymbol{\theta}_w) \| \mathcal{N}(\mathbf{0}, \mathbf{I_d})) - \beta_{\mathbf{l_2}} \|\boldsymbol{\theta_w}\|_2^2$$

**Training.** For each example, the adversary takes as input: (i) the last layer output of the current learner's feature network; and (ii) the input label. The adversary then outputs a weight (in $[0,1]$). The idea of applying the adversary directly on the learner's features (instead of the original input) is based on recent literature (Rosenfeld et al., 2022; Kirichenko et al., 2022) that suggests re-training the prediction head is sufficient for robustness to shifts. The adversary tries to maximize weights on examples with value $\geq \eta$ (hyperparameter) and minimize on others. For the learner, in addition to the example it takes as input the adversary assigned weight for that example from the previous round and uses it to reweigh its loss in a minibatch. Both players are updated in a round (Algorithm 1).

## 5 THEORETICAL ANALYSIS

The main objective of our analysis of BR-DRO is to show how adding a bitrate constraint on the adversary can: (i) give us tighter statistical estimates of the worst risk; and (ii) control the pessimism (excess risk) of the learned solution. First, we provide worst risk generalization guarantees using the PAC-Bayes framework (Catoni, 2007), along with a result for kernel adversary. Then, we provide convergence rates and pessimism guarantees for the solution found by our online solver for a specific instance of $\mathcal{W}(\gamma)$. For both these, we analyze the constrained form of the conditional value at risk (CVaR) DRO objective (Levy et al., 2020) below.

**Bitrate-Constrained CVaR DRO.** When the uncertainty set $\mathcal{Q}$ is defined by the set of all distributions $Q$ that have bounded likelihood *i.e.,* $\|q/p\|_\infty \leq 1/\alpha_0$, we recover the original CVaR DRO

---

[2]We use $\theta_h, \theta_w$ and $l(\theta_h)$ to denote $w(\boldsymbol{\theta}_w; (\mathbf{x}, \mathbf{y})), h(\boldsymbol{\theta}_h; \mathbf{x})$ and $l(h(\boldsymbol{\theta}_h; \mathbf{x}), \mathbf{y})$ respectively.

objective (Duchi & Namkoong, 2021). The bitrate-constrained version of CVaR DRO is given in Equation 6 (see Appendix C for derivation). Note that, slightly different from Section 3, we define $\mathcal{W}$ as the set of all measurable functions $w: \mathcal{X} \times \mathcal{Y} \mapsto [0,1]$, since the other convex restrictions in Equation 1 are handled by dual variable $\eta$. As in Section 4, $\mathcal{W}(\gamma)$ is derived from $\mathcal{W}$ using Definition 4.1. In Equation 6, if we replace the bitrate-constrained class $\mathcal{W}(\gamma)$ with the unrestricted $\mathcal{W}$ then we recover the variational form of unconstrained CVaR DRO in Duchi et al. (2016).

$$\mathcal{L}^*_{\text{cvar}}(\gamma) = \inf_{h \in \mathcal{H}, \eta \in \mathbb{R}} \sup_{w \in \mathcal{W}(\gamma)} R(h,\eta,w) \quad \text{where,} \quad R(h,\eta,w) = (1/\alpha_0)\langle l(h) - \eta, w \rangle_P + \eta \tag{6}$$

**Worst risk estimation bounds for `BR-DRO`.** Since we are only given a finite sampled dataset $\mathcal{D} \sim P^n$, we solve the objective in Equation 6 using the empirical distribution $\hat{P}_n$. We denote the plug-in estimates as $\hat{h}^\gamma_D, \hat{\eta}^\gamma_D$. This incurs an estimation error for the true worst risk. But when we restrict our adversary to $\Delta(\mathcal{W},\gamma)$, for a fixed learner $h$ we reduce the worst-case risk estimation error which scales with the bitrate $\text{KL}(\cdot \| \pi)$ of the solution (deviation from prior $\pi$). Expanding this argument to every learner in $\mathcal{H}$, with high probability we also reduce the estimation error for the worst risk of $\hat{h}^\gamma_D$. Theorem 5.1 states this generalization guarantee more precisely.

**Theorem 5.1** (worst-case risk generalization). *With probability $\geq 1 - \delta$ over $\mathcal{D} \sim P^n$, the worst bitrate-constrained $\alpha_0$-CVaR risk for $\hat{h}^\gamma_D$ can be upper bounded by the following oracle inequality:*

$$\sup_{w \in \mathcal{W}(\gamma)} R(\hat{h}^\gamma_D, \hat{\eta}^\gamma_D, w) \lesssim \mathcal{L}^*_{cvar}(\gamma) + \frac{M}{\alpha_0} \sqrt{\left(\gamma + \log\left(\frac{1}{\delta}\right) + (d+1)\log\left(\frac{L^2 n}{\gamma}\right) + \log n\right)/(2n-1)},$$

*when $l(\cdot,\cdot)$ is $[0,M]$-bounded, L-Lipschitz and $\mathcal{H}$ is parameterized by convex set $\Theta \subset \mathbb{R}^d$.*

Informally, Theorem 5.1 tells us that bitrate-constraint $\gamma$ gracefully controls the estimation error $\mathcal{O}(\sqrt{(\gamma + \mathcal{C}(\mathcal{H}))/n})$ (where $\mathcal{C}(\mathcal{H})$ is a complexity measure) if we know that Assumption 4.2 is satisfied. While this only tells us that our estimator is consistent with $\mathcal{O}_p(1/\sqrt{n})$, the estimate may itself be converging to a degenerate predictor, *i.e.,* $\mathcal{L}^*_{\text{cvar}}(\gamma)$ may be very high. For example, if the adversary can cleanly separate mislabeled points even after the bitrate constraint, then presumably these noisy points with high losses would be the ones mainly contributing to the worst risk, and up-weighting these points would result in a learner that has memorized noise. Thus, it becomes equally important for us to analyze the excess risk (or the pessimism) for the learned solution. Since this is hard to study for any arbitrary bitrate-constrained class $\mathcal{W}(\gamma)$, we shall do so for the specific class of reproducing kernel Hilbert space (RKHS) functions.

**Special case of bounded RKHS.** Let us assume there exists a prior $\Pi$ such that $\mathcal{W}(\gamma)$ in Definition 4.1 is given by an RKHS induced by Mercer kernel $k : \mathcal{X} \times \mathcal{X} \mapsto \mathbb{R}$, s.t. the eigenvalues of the kernel operator decay polynomially, *i.e.,* $\mu_j \lesssim j^{-2/\gamma}$ $(\gamma < 2)$. Then, if we solve for $\hat{h}^\gamma_D, \hat{\eta}^\gamma_D$ by doing kernel ridge regression over norm bounded $(\|f\|_{\mathcal{W}(\gamma)} \leq B \leq 1)$ smooth functions $f$ then we can control: (i) the pessimism of the learned solution; and (ii) the generalization error (Theorem 5.2). Formally, we refer to pessimism for estimates $\hat{h}^\gamma_D, \hat{\eta}^\gamma_D$ as excess risk defined as:

$$\text{excess risk} := \sup_{w \in \mathcal{W}(\gamma)} |\inf_{h,\eta} R(h,\eta,w) - R(\hat{h}^\gamma_D, \hat{\eta}^\gamma_D, w)|. \tag{7}$$

**Theorem 5.2** (bounded RKHS). *For $l, \mathcal{H}$ in Theorem 5.1, and for $\mathcal{W}(\gamma)$ described above $\exists \gamma_0$ s.t. for all sufficiently bitrate-constrained $\mathcal{W}(\gamma)$ i.e., $\gamma \leq \gamma_0$, w.h.p. $1 - \delta$ worst risk generalization error is $\mathcal{O}\big((1/n)\big(\log(1/\delta) + (d+1)\log(nB^{-\gamma}L^{\gamma/2})\big)\big)$ and the excess risk is $\mathcal{O}(B)$ for $\hat{h}^\gamma_D, \hat{\eta}^\gamma_D$ above.*

Thus, in the setting described above we have shown how bitrate-constraints given indirectly by $\gamma, R$ can control both the pessimism and statistical estimation errors. Here, we directly analyzed the estimates $\hat{h}^\gamma_D, \hat{\eta}^\gamma_D$ but did not describe the specific algorithm used to solve the objective in Equation 6 with $\hat{P}_n$. Now, we look at an iterative online algorithm to solve the same objective and see how bitrate-constraints can also influence convergence rates in this setting.

**Convergence and excess risk analysis for an online solver.** In the following, we provide an algorithm to solve the objective in Equation 6 and analyze how bitrate-constraint impacts the solver and the solution. For convex losses, the min-max objective in Equation 6 has a unique solution and this matches the unique Nash equilibrium for the generic online algorithm (game) we describe (Lemma 5.3). The algorithm is as follows: Consider a two-player zero-sum game where the learner uses a no-regret strategy to first play $h \in \mathcal{H}, \eta \in \mathbb{R}$ to minimize $\mathbb{E}_{w \sim \delta} R(h,\eta,w)$. Then, the adversary plays follow the regularized leader (FTRL) strategy

| Method | Waterbirds | | CelebA | | CivilComments | |
|---|---|---|---|---|---|---|
| | Avg | WG | Avg | WG | Avg | WG |
| ERM | 97.1 (0.1) | 71.0 (0.4) | 95.4 (0.2) | 46.9 (1.0) | 92.3 (0.2) | 57.2 (0.9) |
| LfF (Nam et al., 2020) | 90.7 (0.2) | 77.6 (0.5) | 85.3 (0.2) | 77.4 (0.7) | 92.4 (0.1) | 58.9 (1.1) |
| RWY (Idrissi et al., 2022) | 93.7 (0.3) | 85.8 (0.5) | 84.9 (0.2) | 80.4 (0.3) | 91.7 (0.2) | 67.7 (0.7) |
| JTT (Liu et al., 2021) | 93.2 (0.2) | 86.6 (0.4) | 87.6 (0.2) | 81.3 (0.5) | 90.8 (0.3) | 69.4 (0.8) |
| CVaR DRO (Levy et al., 2020) | 96.3 (0.2) | 75.5 (0.4) | 82.2 (0.3) | 64.7 (0.6) | 92.3 (0.2) | 60.2 (0.8) |
| BR-DRO (VIB) (ours) | 94.1 (0.2) | 86.3 (0.3) | 86.7 (0.2) | 80.9 (0.4) | 90.5 (0.2) | 68.7 (0.9) |
| BR-DRO ($l_2$) (ours) | 93.8 (0.2) | 86.4 (0.3) | 87.7 (0.3) | 80.4 (0.6) | 91.0 (0.3) | 68.9 (0.7) |
| Group DRO Sagawa et al. (2019) | 93.2 (0.3) | 91.1 (0.3) | 92.3 (0.3) | 88.4 (0.6) | 88.5 (0.3) | 70.0 (0.5) |

Table 1: **BR-DRO recovers worst group performance gap between CVaR DRO and Group DRO:** On Waterbirds, CelebA and CivilComments we report test average (Avg) and test worst group (WG) accuracies for BR-DRO and baselines. In (·) we report the standard error of the mean accuracy across five runs.

to pick distribution $\delta \in \Delta(\mathcal{W}(\gamma))$ to maximize the same. Our goal is to analyze the bitrate-constraint $\gamma$'s effect on the above algorithm's convergence rate and the pessimistic nature of the solution found. For this, we need to first characterize the bitrate-constraint class $\mathcal{W}(\gamma)$. If we assume there exists a prior $\Pi$ such that $\mathcal{W}(\gamma)$ is Vapnik-Chervonokis (VC) class of dimension $O(\gamma)$, then in Theorem 5.4, we see that the iterates of our algorithm converge to the equilibrium (solution) in $\mathcal{O}(\sqrt{\gamma \log n / T})$ steps. Clearly, the degree of bitrate constraint can significantly impact the convergence rate for a generic solver that solves the constrained DRO objective. Theorem 5.4 also bounds the excess risk (Equation 7) on $\hat{P}_n$.

**Lemma 5.3** (Nash equilibrium). *For strictly convex $l(h)$, $l(h) \in [0, M]$, the objective in Equation 6 has a unique solution which is also the Nash equilibrium of the game above when played over compact sets $\mathcal{H} \times [0, M]$, $\Delta(\mathcal{W}, \gamma)$. We denote this equilibrium as $h_D^*(\gamma), \eta_D^*(\gamma), \delta_D^*(\gamma)$.*

**Theorem 5.4.** *At time step $t$, if the learner plays $(h_t, \eta_t)$ with no-regret and the adversary plays $\delta_t$ with FTRL strategy that uses a negative entropy regularizer on $\delta$ then average iterates $(\bar{h}_T, \bar{\eta}_T, \bar{\delta}_T) = (1/T) \sum_{t=1}^{T} (h_t, \eta_t, \delta_t)$ converge to the equilibrium $(h_D^*(\gamma), \eta_D^*(\gamma), \delta_D^*(\gamma))$ at rate $\mathcal{O}(\sqrt{\gamma \log n / T})$. Further the excess risk defined above is $\mathcal{O}((M/\alpha_0)(1 - \frac{1}{n^\gamma}))$.*

## 6 EXPERIMENTS

Our experiments aim to evaluate the performance of BR-DRO and compare it with ERM and group shift robustness methods that do not require group annotations for training examples. We conduct empirical analyses along the following axes: (i) worst group performance on datasets that exhibit known spurious correlations; (ii) robustness to random label noise in the training data; (iii) average performance on hybrid covariate shift datasets with unspecified groups; and (iv) accuracy in identifying minority groups. See Appendix B for additional experiments and details[3].

**Baselines.** Since our objective is to be robust to group shifts without group annotations on training examples, we explore baselines that either optimize for the worst minority group (CVaR DRO (Levy et al., 2020)) or use training losses to identify specific minority points (LfF (Nam et al., 2020), JTT (Liu et al., 2021)). Group DRO (Sagawa et al., 2019) is treated as an oracle. We also compare with the simple re-weighting baseline (RWY) proposed by Idrissi et al. (2022).

**Implementation details.** We train using Resnet-50 (He et al., 2016) for all methods and datasets except CivilComments, where we use BERT (Wolf et al., 2019). For our VIB adversary, we use a 1-hidden layer neural network encoder and decoder (one for each label). As mentioned in Section 4, the adversary takes as input the learner model's features and the true label to generate weights. All implementation and design choices for baselines were adopted directly from Liu et al. (2021); Idrissi et al. (2022). We provide model selection methodology and other details in Appendix B.

**Datasets.** For experiments in the known groups and label noise settings we use: (i) Waterbirds (Wah et al., 2011) (background is spurious), CelebA (Liu et al., 2015) (binary gender is spuriously correlated with label "blond"); and CivilComments (WILDS) (Borkan et al., 2019) where the task is to predict "toxic" texts and there are 16 predefined groups Koh et al. (2021). We use FMoW and Camelyon17 (Koh et al., 2021) to test methods on datasets that do not have explicit group shifts. In FMoW the task is to predict land use from satellite images where the training/test set comprises of data before/after 2013. Test involves both subpopulation shifts over regions (*e.g.,* Africa, Asia) and domain generalization over time (year). Camelyon17 presents a domain generalization problem where the task is to detect tumor in tissue slides from different sets of hospitals in train and test sets.

---

[3]The code used in our experiments can be found at https://github.com/ars22/bitrate_DRO.

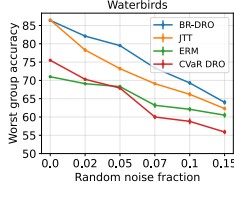 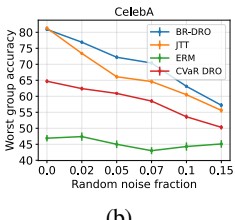 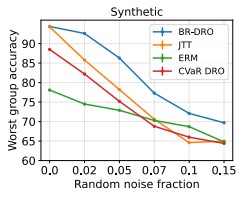

(a)                               (b)

Figure 2: *(Left)* **Visualization (2d) of noisy synthetic data and learned predictors:** We plot the decision boundaries (projected onto core and spurious features) learned by JTT and `BR-DRO` when the adversary is restricted to a sparse predictor. While our method recovers the core feature the baselines memorize the minority points. *(Right)* **`BR-DRO` is robust to random label noise in training data:** Across varying levels of noise fraction in training data we compare performance of `BR-DRO` with ERM and methods (JTT, CVaR DRO) that naively up weight high loss points.

### 6.1 IS `BR-DRO` ROBUST TO GROUP SHIFTS WITHOUT TRAINING DATA GROUP ANNOTATIONS?

Table 1 compares the average and worst group accuracy for `BR-DRO` with ERM and four group shift robustness baselines: JTT, LtF, SUBY, and CVaR DRO. First, we see that unconstrained CVaR DRO underperforms other heuristic algorithms. This matches the observation made by Liu et al. (2021). Next, we see that adding bitrate constraints on the adversary via a KL term or $l_2$ penalty significantly improves the performance of `BR-DRO` (VIB) or `BR-DRO` ($l_2$), which now matches the best performing baseline (JTT). Thus, we see the less conservative nature of `BR-DRO` allows it to recover a large portion of the performance gap between Group DRO and CVaR DRO. Indirectly, this partially validates our Assumption 4.2, which states that the minority group is identified by a low bitrate adversary class. In Section 6.3 we discuss exactly what fraction of the minority group is identified, and the role played by the strength of bitrate-constraint.

### 6.2 `BR-DRO` IS MORE ROBUST TO RANDOM LABEL NOISE

Several methods for group robustness (*e.g.,* CVaR DRO, JTT) are based on the idea of up weighting points with high training losses. The goal is to obtain a learner with matching performance on every (small) fraction of points in the dataset. However, when training data has mislabeled examples, such an approach will likely yield degenerate solutions. This is because the adversary directly upweights any example where the learner has high loss, including datapoints with incorrect labels. Hence, even if the learner's prediction matches the (unknown) true label, this formulation would force the learner to memorize incorrect labelings at the expense of learning the true underlying function. On the other hand, if the adversary is sufficiently bitrate constrained, it cannot upweight the arbitrary set of randomly mislabeled points, as this would require it to memorize those points. Our Assumption 4.2 also dictates that the distribution shift would not upsample such high bitrate noisy examples. Thus, our constraint on the adversary ensures `BR-DRO` is robust to label noise in the training data and our assumption on the target distribution retains its robustness to test time distribution shifts.

In Figure 2b we highlight this failure mode of unconstrained up-weighting methods in contrast to `BR-DRO`. We first induce random label noise (Carlini et al., 2019) of varying degrees into the Waterbirds and CelebA training sets. Then we run each method and compare worst group performance. In the absence of noise we see that the performance of JTT is comparable with `BR-DRO`, if not slightly better (Table 1). Thus, both `BR-DRO` and JTT perform reasonably well in identifying and upsampling the simple minority group in the absence of noise. In its presence, `BR-DRO` significantly outperforms JTT and other approaches on both Waterbirds and CelebA, as it only upsamples the minority examples misclassified by simple features, ignoring the noisy examples for the reasons above. To further verify our claims, we set up a noisily labeled synthetic dataset (see Appendix B for details). In Figure 2a we plot training samples as well as the solutions learned by `BR-DRO` and and JTT on synthetic data. In Figure 1(*right*) we also plot exactly which points are upweighted by `BR-DRO` and JTT. Using both figures, we note that JTT mainly upweights the noisy points (in red) and memorizes them using $\mathbf{x}_{noise}$. Without any weights on minority, it memorizes them as well and learns component along spurious feature. On the contrary, when we restrict the adversary with `BR-DRO` to be sparse ($l_1$ penalty), it only upweights minority samples, since no sparse predictor can separate noisy points in the data. Thus, the learner can no longer memorize the upweighted minority and we recover the robust predictor along core feature.

### 6.3 WHAT FRACTION OF MINORITY IS RECOVERED BY `BR-DRO`?

We claim that our less pessimistic objective can more accurately recover (upsample) the true minority group if indeed the minority group is simple (see Assumption 4.2 for our definition of simple). In this section, we aim to verify this claim. If we treat examples in the top $10\%$ (chosen for post hoc analysis)

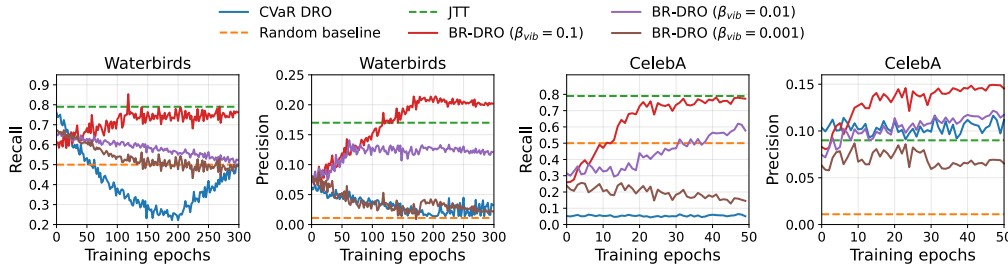

Figure 3: By considering the fraction of points upweighted by our adversary (top 10%) as the positive class we analyze the precision and recall of this class with respect to the minority group. and do the same for JTT, random baseline and CVaR DRO. `BR-DRO` achieves highest precision and matches recall with JTT asymptotically. We also find that increasing bitrate constraint $\beta_{vib}$ helps improving precision/recall.

fraction of examples as our predicted minorities, we can check precision and recall of this decision on the Waterbirds and CelebA datasets. Figure 3 plots these metrics at each training epoch for `BR-DRO` (with varying $\beta_{vib}$), JTT and CVaR DRO. Precision of the random baseline tells us the true fraction of minority examples in the data. First we note that `BR-DRO` consistently performs much better on this metric than unconstrained CVaR DRO. In fact, as we reduce strength of $\beta_{vib}$ we recover precision/recall close to the latter. This controlled experiment shows that the bitrate constraint is helpful (and very much needed) in practice to identify rare simple groups. In Figure 3 we observe that asymptotically, the precision of `BR-DRO` is better than JTT on both datasets, while the recall is similar. Since importance weighting has little impact in later stages with exponential tail losses (Soudry et al., 2018; Byrd & Lipton, 2019), other losses (*e.g.*, polytail Wang et al. (2021)) may further improve the performance of `BR-DRO` as it gets better at identifying the minority classes when trained longer.

### 6.4    How does `BR-DRO` perform on more general covariate shifts?

In Table 2 we report the average test accuracies for `BR-DRO` and baselines on the hybrid dataset FMoW and domain generalization dataset Camelyon17. Given its hybrid nature, on FMoW we also report worst region accuracy. First, we note that on these datasets group shift robustness baselines do not do better than ERM. Some are either too pessimistic (*e.g.,* CVaR DRO), or require heavy assumptions (*e.g.,* Group DRO) to be robust to domain generalization. This is also noted by Gulrajani & Lopez-Paz (2020). Next, we see that `BR-DRO` ($l_2$ version) does better than other group shift baselines on both both worst region and average datasets and matches ERM performance on Camelyon17. One explanation could be that even though these datasets test models on new domains, there maybe some latent groups defining these domains that are simple and form a part of latent subpopulation shift. Investigating this claim further is a promising line of future work.

| Method | FMoW | | Camelyon17 |
|---|---|---|---|
| | Avg | W-Reg | Avg |
| ERM | 53.3 (0.1) | 32.4 (0.3) | 70.6 (1.6) |
| JTT Liu et al. (2021) | 52.1 (0.1) | 31.8 (0.2) | 66.3 (1.3) |
| LfF Nam et al. (2020) | 49.6 (0.2) | 31.0 (0.3) | 65.8 (1.2) |
| RWY Idrissi et al. (2022) | 50.8 (0.1) | 30.9 (0.2) | 69.9 (1.3) |
| Group DRO Sagawa et al. (2019) | 51.9 (0.2) | 30.4 (0.3) | 68.5 (0.9) |
| CVaR DRO Levy et al. (2020) | 51.5 (0.1) | 31.0 (0.3) | 66.8 (1.3) |
| `BR-DRO` (VIB) (ours) | 52.0 (0.2) | 31.8 (0.2) | 70.4 (1.5) |
| `BR-DRO` ($l_2$) (ours) | 53.1 (0.1) | 32.3 (0.2) | 71.2 (1.0) |

Table 2:  Average (Avg) and worst region (W-Reg for FMoW) test accuracies on Camelyon17 and FMoW.

## 7    Conclusion

In this paper, we proposed a method for making machine learning models more robust. While prior methods optimize robustness on a per-example or per-group basis, our work focuses on features. In doing so, we avoid requiring group annotations on training samples, but also avoid the excessively conservative solutions that might arise from CVaR DRO with fully unconstrained adversaries. Our results show that our method avoids learning spurious features, is robust to noise in the training labels, and does better on other forms of covariate shifts compared to prior approaches. Our theoretical analysis also highlights other provable benefits in some settings like reduced estimation error, lower excess risk and faster convergence rates for certain solvers.

**Limitations.** While our method lifts the main limitation of Group DRO (access to training group annotations), it does so at the cost of increased complexity. Further, to tune hyperparameters, like prior work we assume access to a some group annotations on validation set but also get decent performance (on some datasets) with only a balanced validation set (see Appendix B). Adapting group shift methods to more generic settings remains an important and open problem.

**Acknowledgement.** The authors would like to thank Tian Li, Saurabh Garg at Carnegie Mellon University, and Yoonho Lee at Stanford University for helpful feedback and discussion.

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
