# OpenReview forum: "Bitrate-Constrained DRO: Beyond Worst Case Robustness To Unknown Group Shifts"
_ICLR.cc/2023/Conference — ICLR 2023 poster_

### Official Review · Reviewer_WsDK · 2022-10-24

**Confidence:** 3
**Clarity, Quality, Novelty And Reproducibility:** This paper is well written.
**Correctness:** 4
**Technical Novelty And Significance:** 2
**Empirical Novelty And Significance:** 3
**Recommendation:** 6

**Strength And Weaknesses:**

Strength: This "simple group shifted" phenomenon is well-motivated and using this well-defined, easy-to-solve DRO to capture such phenomena without knowing group information is a clean framework and has been well-supported by experiments. (But I am not an empirical person so my evaluation of empirical results has low confidence).

Weakness:
1\ All the theorem seem a bit incremental. Theorem 5.1 is a classical generalization bound analysis and 5.2 is s RKHS-specialized version. Theorem 5.4 highly relies on the assumption that $W(\gamma)$ is Vapnik-Chervenokis (VC) class of dimension $O(\gamma)$ and other standard techniques from  Abernethy et al. (2018) . But I think it is ok given it is not a purely theoretical paper.
2\ To me, the connection between "simple group shift" and "bitrate-constrained distribution" is a bit weak. Seems to me the group shift can violate the bitrate-constrained distribution and the bitrate-constrained distribution can also imply a non-group shift. Of course, I think the explanation from the paper is a good intuition...



**Summary Of The Paper:**

This paper proposes bitrate-constrained DRO and makes a connection with the "simple group shift" conception. Specifically, they argue that this constrained DRO can distinguish between a rare group that is realized by simple functions and a rare group of examples that share no feature in common or may even be mislabeled, without knowing the group information in advance, and can thus improve the performance of neural networks by avoiding being too pessimistic. To support their result, they both give a theoretical generalization guarantee as well as many empirical experiments. In addition, they proposed a modified online game-based strategy to solve this optimization problem.

**Summary Of The Review:**

This paper to me is slightly above the acceptance borderline because it provides a clean and well-defined framework, a good intuitive explanation, and convincing experimental results. But like I said above, its theoretical results are still relatively incremental and their justification on the connection between  "simple group shift" and "bitrate-constrained distribution" is a bit weak.

---

> ### Author Response · Authors · 2022-11-17
> **Response to Reviewer WsDK**
>
> We thank the reviewer for the thorough review, and the detailed suggestions for improving the paper. We hope the clarifications that follow address all of the concerns in the review. Please let us know if any issues remain, or if all the issues have been addressed!
>
> > All the theorem seem a bit incremental…. But I think it is ok given it is not a purely theoretical paper.
>
> We agree that the main contributions of this paper are empirical, though emphasize that similar prior methods often lack theoretical guarantees (e.g., the JTT, LfF or the debiasing baseline BPA suggested by reviewer BZ25). That being said, we hope our theoretical results (which shows how the bitrate-constraint controls the statistical efficiency and excess risk) can offer some useful insight into the empirical benefits we observe when using BR-DRO in practice.
>
> > Connection between "simple group shift" and ''bitrate-constrained distribution'' ... Seems to me the group shift can violate the bitrate-constrained distribution and the bitrate-constrained distribution can also imply a non-group shift.
>
>
> We formally define what we mean by ''simple group shift'' in Assumption 4.2, which is closely tied to the definition of a ''bitrate-constrained'' class in Definition 4.1. Since each group $G$ in the group structure $\mathcal{G}(P, Q_0)$ for source $P$ and target $Q_0$ (Definition 3.1) can be equivalently represented by the indicator function $w_G = \mathbb{I}((\mathbf{x}, y) \in G)$, we characterize a group shift as a simple group shift if the function $w_G$  $(\forall G \in \mathcal{G}(P, Q_0))$ is contained in a bitrate-constrained class $\mathcal{W}(\gamma)$. As we mention after Definition 4.1 in Section 4, $\mathcal{W}(\gamma)$ is bitrate-constrained since it only contains means of those distributions (over functions) that do not deviate from prior $\pi$ under the KL constraint and thus can be described with fewer bit rates given $\pi$. To clarify this connection, we have revised the paper by **adding a practical example in Appendix A.3**.
>
> Yes, we agree with the reviewer that in general group shifts can violate the bitrate-constraint assumption $-$ those are precisely the ones we characterize as ''not simple''. Also, it is true that bitrate-constrained re-weighting functions can also imply non-group shifts (violating Definition 3.1), for e.g., when absolute continuity is violated $(Q_0 \centernot{\ll} P)$. In this work, we only focus on distribution shifts satisfying Assumption 3.3 (standard group shift) which is fairly generic, and use the bitrate-constrained class of re-weighting functions to characterize the group shift as ''simple'' or ''not simple''.

---

> > ### Author Response · Authors · 2022-11-28
> > **Are there any additional questions or concerns?**
> >
> > We hope that our responses above have addressed the concerns raised in the review. **Are there any additional concerns? We would be happy to continue the discussion if there are additional questions or concerns.**

---

### Official Review · Reviewer_BZ25 · 2022-10-24

**Confidence:** 3
**Correctness:** 3
**Technical Novelty And Significance:** 3
**Empirical Novelty And Significance:** 2
**Recommendation:** 6

**Clarity, Quality, Novelty And Reproducibility:**

- The paper writing is well structured.
- This work nicely bridges the gap between unconstrained DRO and group DRO with a practical assumption in a novel way.
- Source codes are provided for reproducibility.

**Strength And Weaknesses:**

Strength
- The paper address why restricted adversary for DRO do better than unconstrained one, which is supported by theoretical justification under the assumption of simple group shift.
- The proposed BR-DRO framework is empirically validated by extensive experiments.

Weakness
- Several important unsupervised comparisons are missing. Especially, [George] and [BPA] extend Group DRO in an unsupervised way, where they conduct clustering on the feature space and identify latent subclass (minority group). In other words, they also aim to identify potential groups for worst-case robustness without group annotations.
- The formulation of BR-DRO starts from Assumption 4.2, but what is the motivation of the assumption? Also, I think it may be applied well because most of group robustness datasets has binary (bias) attributes, Does this simple group shift assumption still work well in datasets where the bias attribute has multiple values?

[George] No Subclass Left Behind: Fine-Grained Robustness in Coarse-Grained Classification Problems, NeurIPS 2020

[BPA] Unsupervised Learning of Debiased Representations with Pseudo-Attributes, CVPR 2022

**Summary Of The Paper:**

Distributionally robust optimization methods are often overly convservative, while group DRO approaches requires group annotations, which limits its practicality. To address this challenge, they introduce a practical assumption that group shift is captured by high-level features with a simple function. Based on this assumption, the authors propose a BR-DRO, which makes the model to be robust against distribution shifts along group which is realized by simple functions. This helps model not to be overly conservative while being robust to plausible worst cases, resulting in meaningful performance improvement compared to unconstrained DRO.

**Summary Of The Review:**

The authors constrains the distributionally robust optimization method under the assumption of group shift, which prevents the model from being overly conservative and pessimistic. The practical relaxation of BR-DRO is novel and well supported by experimental results, though those are not that strong compared to recent debiasing approaches. Additional discussions with other robust optimization approaches can help for better understanding.

---

> ### Author Response · Authors · 2022-11-18
> **Response to Reviewer BZ25**
>
> We thank the reviewer for the thorough review, and the detailed suggestions for improving the paper. We have added Appendix B.6 which compares BR-DRO to recent clustering dependent debiasing approaches suggested by the reviewer: BPA and George. Additionally, using the example of CivilComments dataset we have added discussion on BR-DRO's robustness to spurious attributes with ''multiple values'' in Appendix B.5. We hope these modifications and clarifications address all of the concerns in the review. Please let us know if any issues remain, or if all the issues have been addressed!
>
> > Several important unsupervised comparisons are missing. Especially, [George] and [BPA] extend Group DRO in an unsupervised way, where they conduct clustering on the feature space and identify latent subclass (minority group). In other words, they also aim to identify potential groups for worst-case robustness without group annotations.
>
> As suggested by the reviewer, we compare BR-DRO to George and BPA on Waterbirds, CelebA and FMoW (results and discussion have been added to appendix B.6). On the worst group performance, we find that BR-DRO does better than George on all datasets. On the same metric, BR-DRO also does better than BPA on Waterbirds and FMoW, while being comparable to it on CelebA. Unlike our method, both baselines have a stagewise training procedure while also lacking a clear definition of the robust set (that we outline in our Assumption 4.2). Since both these methods do not up-weight arbitrary points with high losses, we can think of them as having an implicit constraint on their weighting schemes (adversary), thus possibly yielding solutions that are less pessimistic than CVaR DRO. At the same time, they lack excess risk (pessimism) guarantees (that we have in Section 5), for their learned solutions.
>
> > The formulation of BR-DRO starts from Assumption 4.2, but what is the motivation of the assumption? Also, I think it may be applied well because most of group robustness datasets has binary (bias) attributes, Does this simple group shift assumption still work well in datasets where the bias attribute has multiple values?
>
> If we do not make the simple group shift assumption in Assumption 4.2, then the DRO formulation would try to optimize for performance on contrived shifts, where the adversary can pick an arbitrary measurable set to upweight (2nd paragraph in Section 4). Prior work [a] tells us that being robust to unconstrained adversaries can lead to pessimistic solutions that guard against unrealistic shifts as opposed to caring for realistic ones along meaningful degrees of variation (e.g., background, color, orientation) which is more reflective of practical scenarios (e.g., the Waterbirds example in Section 1, Figure 1). Further, we clarify in Section 4 that: (i) neural networks tend to perform poorly on minority groups that are simple (partly attributed to the simplicity bias of neural nets [b]); and (ii) the bitrate-constraint on the adversary helps identify the worst minority group that is simple, motivating both our Assumption 4.2 and the BR-DRO objective in Equation 4.
>
> The simple group shift assumption works even when the spurious attribute is not binary as long as the group $G$ corresponds to an indicator function $\mathbb{I}(\mathbf{x}, y \in G)$ (e.g., an intersection of hyperplanes) that is realized in a low-bitrate class (e.g., a neural net with a VIB constraint which is how we model the adversary in the VIB version of BR-DRO). We thank the reviewer for this question and run additional evaluations to verify the above on the CivilComments dataset where there are 8 binary spurious attributes. While conventionally [c] the evaluation on this dataset reports worst-performance across $2  \times 8$  groups, we add results in Appendix B.5 where we additionally evaluate worst-performance across $2 \times ^8C_2$ groups – i.e. combinations of binary attributes (making the spurious attribute categorical). We find that the performance of VIB BR-DRO (62.9%) is still significantly better than the unconstrained CVaR DRO (56.5%), while matching the oracle Group DRO.
>
> [a] Does distributionally robust supervised learning give robust classifiers?: W Hu, G Niu, I Sato… - … Conference on Machine …, 2018 - proceedings.mlr.press
>
> [b] Harshay Shah, Kaustav Tamuly, Aditi Raghunathan, Prateek Jain, and Praneeth Netrapalli. The pitfalls of simplicity bias in neural networks. Advances in Neural Information Processing Systems, 33:9573–9585, 2020.
>
> [c] Wilds: A benchmark of in-the-wild distribution shifts PW Koh, S Sagawa, H Marklund… - International …, 2021 - proceedings.mlr.press

---

> > ### Author Response · Authors · 2022-11-28
> > **Are there any additional questions or concerns?**
> >
> > We hope that the revisions in Appendix B.6 (comparisons with BPA, George) and Appendix B.5 (an example of multi-valued spurious attribute), along with other responses above have addressed the concerns raised in the review. **Are there any additional concerns? We would be happy to continue the discussion if there are additional questions or concerns.**

---

> > > ### Comment · Reviewer_BZ25 · 2022-12-01
> > > **Response to authors**
> > >
> > > Thanks the authors for additional experiments and analysis regarding my concerns, most of which are well addressed. I will keep my original rating as is.

---

### Official Review · Reviewer_EndR · 2022-10-24

**Confidence:** 3
**Correctness:** 4
**Technical Novelty And Significance:** 3
**Empirical Novelty And Significance:** 3
**Recommendation:** 8

**Clarity, Quality, Novelty And Reproducibility:**

Clarity: I think this paper is well-written.

Quality: I think this paper is good and vote to accept it.

Novelty: I think the BR-DRO constraint is novel and elegant.

Reproducibility: The authors provide code in the supplementary material, but I have not run it to verify their claims. They give a detailed description of their experiments in the `Experiments' section of their paper.

**Strength And Weaknesses:**

Strengths:
1. BR-DRO has a well-motivated constraint and is amenable to clean theoretical analysis.
2. BR-DRO performs well on the benchmark datasets and recovers the kind of guarantees GROUP-DRO achieves without using the group information.
3. I think the paper is well-written and clear.

Weaknesses:
1. As expected, BR-DRO has higher complexity when compared to GROUP-DRO.

**Summary Of The Paper:**

This paper proposes a new method for the problem of distributionally robust optimization. The learning procedure ‘Bitrate-Constrained DRO’ (BR-DRO) provides robustness to distribution shifts along groups realised by simple functions. This procedure is able to match the performance of methods that use group information on training samples, despite not actually using this information.

In comparison to prior works such as DRO, which generally optimizes for worst-case performance on distributions that lie in an f-divergence ball around the training distribution, BR-DRO further constrains the adversary’s class, while not requiring more information about the underlying group structure. Other approaches to constrain the adversary, the resulting set seems to still contain some mislabelled and hard instances from the majority set. Other works which seem to obtain group-robustness without group information either implicitly assume the group structure, or are vulnerable to label noise.

Their main theoretical result is an upper bound on the estimation error of the BR-DRO estimator in terms of the KL divergence from the prior.

They provide experiments comparing BR-DRO to ERM and other group shift robustness methods that do not require annotations for training samples.

**Summary Of The Review:**

I think this paper is good and vote to accept it. It introduces a novel constrained version of DRO which is performs as well as GROUP-DRO without the group information. The paper is well-written and provides experiments supporting this choice of modelling assumptions.

One interesting modelling assumption on the adversary they introduce, is that the adversary can only separate datapoints into potential groups using simple features.

---

> ### Author Response · Authors · 2022-11-17
> **Response to Reviewer EndR**
>
> We thank the reviewer for the thorough review. The reviewer is correct that BR-DRO adds statistical and computational complexity relative to Group-DRO, though we also see BR-DRO as a more practical approach as it loosens the requirement of group annotations while retaining competitive performance. Please let us know if you have any additional questions or if all the issues have been addressed!

---

### Official Review · Reviewer_Pjrc · 2022-10-31

**Confidence:** 3
**Correctness:** 3
**Technical Novelty And Significance:** 3
**Empirical Novelty And Significance:** 2
**Recommendation:** 5

**Clarity, Quality, Novelty And Reproducibility:**

Clarity:The clarity of the paper can be improved relatively easily, in my opinion, so I will wait for the rebuttal/response of authors.

Quality:The paper is high quality

Reproducibility: I have not thoroughly checked the code, but authors have provided detailed code to reproduce all results, included datasets wherever necessary.

**Strength And Weaknesses:**

Strengths:
1. The paper is largely well written and motivated.
2. The problem setup is well motivated and being robust to unknown but well modeled group shift is an interesting approach to address lack of robustness of ML models to spurious correlations (if modeled as potential groups).

Weaknesses:
Although the paper is well written, there are many clarifications necessary that could improve the paper:
1. I am guessing $z$ in Equation 5 corresponds to latent group identities? It seems to be introduced in a rush without explaining the objective function at all, leaving it to the reader to decipher details.

2. Unclear motivation of why adversary needs to be a deep neural network and then add l-2 regularization to the parametrization. I would've liked to see a simpler set up where an exact solution is available, and then a heuristic proposed when adversary is also a complex model. It is also not clear how this ties to the assumption that the group identity function is actually a simple function.

3. The latent group identification via KL regularization is interesting, but then I am a little confused what happens if the group identities somehow overlap with class labels themselves, guessing no real robust learning can happen then?

4. Comparison of Theorem 5.1 to vanilla group DRO would be more informative.

5. Same overall concern: What is the applicability of theorem 5.4 to the heuristic assumption of adversary being neural network with regularization added in the objective function.

6. Empirical evaluation could also use more insights:
   6.1 Are celebA results worse compared to other datasets for domain-specific reasons? If yes, why is the worst-group test accuracy lower for Bit-rate DRO always?
   6.2 If the benefit of BR-DRO is more visible only with label noise, how practically useful is BR-DRO?
   6.3 What are the groups used by group DRO vs BR-DRO and how is that affected by the choice of regularization, beyond the fact that higher regularization will constrain the function to be less complex.

**Summary Of The Paper:**

This paper proposes bit-rate constrained group DRO, where the assumption is that group identities can be modeled by a simple function class. This allows the framework to improve the overall utility of the learned model by relying less on arbitrarily chosen mis-labeled points (in a DRO framework), but ensure that only once that are also well modeled by the bit-rate constrained function are upweighted. Using the motivation, an objective is proposed to come up with a local minima where both the adversary and the hypothesis model is a neural network. Theoretical analysis provides risk bounds as well as analysis on convergence for online solver is provided. Experimental results demonstrate some benefit of the approach on real world data, including domain shift benchmarks.

**Summary Of The Review:**

Overall, I am a little unsure about some modeling choices and practical utility of the paper. But I will look forward to the response and update my review accordingly.

---

> ### Author Response · Authors · 2022-11-17
> **Response to Reviewer Pjrc (Part 1)**
>
> We thank the reviewer for the thorough review, and the detailed suggestions for improving the paper. We revise our paper with a full derivation for Equation (5) clarifying the exact parameterization of the adversary in Appendix A.2. We also add a comparison of the result in Theorem 5.1 with Group DRO in Appendix C.6, and answer other questions below. Please let us know if these modifications and clarifications address all of your concerns, or if there are any other issues we can address.
>
> > I am guessing z in Equation 5 corresponds to latent group identities? It seems to be introduced in a rush without explaining the objective function at all, leaving it to the reader to decipher details.
>
> We have appropriately revised Section 4 where we introduce our practical objective in Equation (5) and have also added Section A.2 to discuss the objective in detail. The latent variable $\mathbf{z}$ corresponds to the output of the hidden layer of VIB when the adversary is parameterized as a VIB in our BR-DRO formulation. While we assume the adversary’s action set to be bitrate-constrained in our theoretical analysis in Section 5, in practice we enforce this by adding an information bottleneck penalty (KL constraint) over the representation $\mathbf{z}$. Prior works [a, b] suggest that the information bottleneck is an effective regularizer that would bias the function towards low bitrate solutions that satisfy our Assumption 4.2.
>
> [a] Deep variational information bottleneck: AA Alemi, I Fischer, JV Dillon, K Murphy - arXiv preprint arXiv:1612.00410, 2016 - arxiv.org
>
> [b] Deep learning and the information bottleneck principle: N Tishby, N Zaslavsky - 2015 ieee information theory workshop …, 2015 - ieeexplore.ieee.org
>
>
> > Unclear motivation of why adversary needs to be a deep neural network and then add l-2 regularization to the parametrization. I would've liked to see a simpler set up where an exact solution is available ... not clear how this ties to the assumption that the group identity function is actually a simple function.
>
> Conditioned on the label, the VIB version of BR-DRO parameterizes the adversary as a one hidden layer network with ReLU activations, with an information bottleneck constraint on the hidden layer’s activations. The $l_2$ version of BR-DRO parameterizes the adversary as a single linear transform, with $l_2$ norm constraints on the weights. We do not use deep nets with $l_2$ regularization in any of our experiments. In both versions of BR-DRO, the input to the adversary is: i) the label $y$; and ii) the frozen feature vector that is given by the last layer of the learner’s network (Section 4 BR-DRO training and Appendix A.2).
>
> The information bottleneck constraint in the VIB enforces the assumption we make on the re-weighting function being a low bitrate function [a, b] (Definition 4.1) which directly ties the bottleneck regularizer to our simple group shift assumption in Assumption 4.2. For the $l_2$ version of BR-DRO, the regularizer constraints the $l_2$ norm of the linear layer, which corresponds to bitrate constraints under certain priors [c]. Moreover, the class of linear functions has a bounded VC dimension, making our theoretical results in Section 5 (Theorem 5.4) directly applicable to the $l_2$ version. Perhaps, the $l_2$ version with the linear adversary is closest to the ''simpler set up'' referenced by the reviewer.
>
> [a] Learning deep representations by mutual information estimation and maximization: RD Hjelm, A Fedorov, S Lavoie-Marchildon… - arXiv preprint arXiv …, 2018 - arxiv.org
>
> [b] Deep learning and the information bottleneck principle: N Tishby, N Zaslavsky - 2015 ieee information theory workshop …, 2015 - ieeexplore.ieee.org
>
> [c] Bayesian regularization: From Tikhonov to horseshoe: NG Polson, V Sokolov - Wiley Interdisciplinary Reviews …, 2019 - Wiley Online Library
>
> > The latent group identification via KL regularization is interesting, but then I am a little confused what happens if the group identities somehow overlap with class labels themselves, guessing no real robust learning can happen then?
>
> In the extreme case where the groups are fully defined by the labels, the BR-DRO objective would only enforce fair performance across labels, building robustness to label shifts. Further, it is easy to see that the fully overlapping case satisfies Assumption 4.2, since the function identifying the $k^\textrm{th}$ label (class): $w(\mathbf{x}, y) = \mathbb{I}(y = k)$ is a simple function contained in a low bitrate class. But if the labels merely correlate with the groups (i.e., some but not total overlap), then robust learning is still possible, and indeed this case does fall under our definition of group (Definition 3.1). Specifically, we define the identity of a group by an indicator function over the $\mathcal{X} \times \mathcal{Y}$ space, i.e. a group is defined by both the input, label combination and the same is true for the re-weighting function in Equation 1.

---

> > ### Author Response · Authors · 2022-11-17
> > **Response to Reviewer Pjrc (Part 2)**
> >
> > > Comparison of Theorem 5.1 to vanilla group DRO would be more informative.
> >
> > We thank the reviewer for the suggestion. Since vanilla group DRO assumes group knowledge it is straightforward to apply traditional generalization bounds over each group first and then use a union bound over the $K$ predetermined groups to get the worst-risk generalization bound. We have added a section on this in Appendix C.6. Since there is no bitrate-constraint in Group DRO the generalization bound for the learner no longer depends on $\gamma$, but there is an explicit dependence on the number of groups $K$ in the Hoeffding term $\log(2K/\delta)$. Given that the knowledge of the groups is assumed, the bound is tighter than BR-DRO by at least $\mathcal{O}(\sqrt{(\log n)/n})$ additive term.
> >
> > > Same overall concern: What is the applicability of theorem 5.4 to the heuristic assumption of adversary being neural network with regularization added in the objective function.
> >
> >
> > The goal of Theorem 5.4 is to provide convergence rates (when the adversary plays a specific FTRL strategy) and to bound the excess risk in terms of the bitrate-constraint $\gamma$ (in Definition 4.1). In fact, for the $l_2$ version of BR-DRO the adversary is a linear layer (see Appendix A.2) which has a VC dimension $\Theta(d)$. Thus, setting $\gamma = \Theta(d)$ would make Theorem 5.4 directly applicable for at least the $l_2$ version of BR-DRO.
> >
> >
> > > Empirical evaluation could also use more insights: 6.1 Are celebA results worse compared to other datasets for domain-specific reasons? If yes, why is the worst-group test accuracy lower for Bit-rate DRO always? 6.2 If the benefit of BR-DRO is more visible only with label noise, how practically useful is BR-DRO? 6.3 What are the groups used by group DRO vs BR-DRO and how is that affected by the choice of regularization, beyond the fact that higher regularization will constrain the function to be less complex.
> >
> > 6.1: In Table 1, the worst-group accuracy of BR-DRO is always comparable to the best performing baseline (up to overlapping standard errors). Regarding CelebA, the task involves predicting whether the person has blond hair or not – which can be a harder learning task than those presented by other datasets, explaining the lower average accuracy in general. Our reported accuracies match prior works [Liu et al. 2021, Levy et al. 2020, Idrissi et al. 2022].
> >
> > 6.2: Popular datasets (e.g., Imagenet, Food-101N, WebVision etc.) collected for many practical tasks exhibit label noise [a, b]. Further the size of the minority groups (in datasets with known group annotations) is typically small ($<3$% in Waterbirds, CelebA, MultiNLI). Since this can likely be of the same order as the fraction of label noise in the data, directly deploying methods like JTT or CVaR DRO on real world tasks can lead to arbitrarily bad performance (as confirmed by our preliminary experiments in Section 6.2). Thus, we believe that being robust to label noise is in itself practically very useful. Further, even in settings without label noise, BR-DRO is practically useful as it allows for competitive performance with existing baselines but without requiring group annotations for every training sample.
> >
> > 6.3: Since datasets FMoW and Camelyon17 do not exhibit sub-population shifts, there are no clear groups. In the absence of true groups, we fall back on prior work [c] to train Group DRO using time periods (years) and hospitals as underlying groups for FMoW and Camelyon17 respectively. For datasets CelebA and Waterbirds, where the ground truth groups are known, we quantify what fraction of the true groups are recovered by BR-DRO and the role played by the higher regularization strength (see Section 6.4).
> >
> > [a] Re-labeling imagenet: from single to multi-labels, from global to localized labels: S Yun, SJ Oh, B Heo, D Han… - Proceedings of the …, 2021 - openaccess.thecvf.com
> >
> > [b] Learning with noisy labels revisited: A study using real-world human annotations: J Wei, Z Zhu, H Cheng, T Liu, G Niu, Y Liu - arXiv preprint arXiv …, 2021 - arxiv.org
> >
> > [c] Wilds: A benchmark of in-the-wild distribution shifts PW Koh, S Sagawa, H Marklund… - International …, 2021 - proceedings.mlr.press

---

> > > ### Author Response · Authors · 2022-11-18
> > > **Are there any additional questions or concerns?**
> > >
> > > We hope that the revisions in Appendix A.2 (detailing Equation 5) and Appendix C.6 (comparing Theorem 5.1 to Group DRO), along with other responses above have addressed the concerns raised in the review. **Are there any additional concerns? We would be happy to continue the discussion if there are additional questions or concerns.**

---

### Decision · Program_Chairs · 2023-01-20

**Decision:**

Accept: poster

**Justification For Why Not Higher Score:**

Given that this paper is about a very technical topic and the scores are not very high, I recommend it as a poster.

**Justification For Why Not Lower Score:**

Only one reviewer is giving a score below 6, but the main complaint is about the "practical utility of the paper". Given that this is a mainly theoretical paper, the practical utility should not be the main concern. So, overall I judge this paper as a clear accept.

**Metareview: Summary, Strengths And Weaknesses:**

This paper proposes bitrate-constrained DRO and it makes a connection with the "simple group shift" conception. Theoretical as well as empirical experiments are shown.

All the reviewers agree that the paper is interesting, well-motivated and with a clean theoretical formulation. Hence, I recommend acceptance of this paper.

**Note From Pc:**

if the above contains the word "oral" or "spotlight" please see: "oral" presentation means -> notable-top-5% and "spotlight" means -> notable-top-25%. As stated in our emails, we are disassociating presentation type from AC recommendations